# GmSWEET29 and Paralog GmSWEET34 Are Differentially Expressed between Soybeans Grown in Eastern and Western Canada

**DOI:** 10.3390/plants11182337

**Published:** 2022-09-07

**Authors:** Julia C. Hooker, Nour Nissan, Doris Luckert, Gerardo Zapata, Anfu Hou, Ramona M. Mohr, Aaron J. Glenn, Brent Barlow, Ketema A. Daba, Thomas D. Warkentin, François Lefebvre, Ashkan Golshani, Elroy R. Cober, Bahram Samanfar

**Affiliations:** 1Agriculture and Agri-Food Canada, Ottawa, ON K1A 0C6, Canada; 2Department of Biology, Ottawa Institute of Systems Biology, Carleton University, Ottawa, ON K1S 5B6, Canada; 3Canadian Centre for Computational Genomics, Montréal, QC H3A 0G1, Canada; 4Agriculture and Agri-Food Canada, Morden, MB R6M 1Y5, Canada; 5Agriculture and Agri-Food Canada, Brandon, MB R7A 5Y3, Canada; 6Crop Development Centre, University of Saskatchewan, Saskatoon, SK S7N 5A8, Canada

**Keywords:** *Glycine max*, SWEET, sugar transport, RNA-seq, transcriptomics

## Abstract

Over the past two decades soybeans grown in western Canada have persistently had lower seed protein than those grown in eastern Canada. To understand the discrepancy in seed protein content between eastern- and western-grown soybeans, RNA-seq and differential expression analysis have been investigated. Ten soybean genotypes, ranging from low to high in seed protein content, were grown in four locations across eastern (Ottawa) and western (Morden, Brandon, and Saskatoon) Canada. Differential expression analysis revealed 34 differentially expressed genes encoding *Glycine max* Sugars Will Eventually be Exported Transporters (GmSWEETs), including paralogs GmSWEET29 and GmSWEET34 (AtSWEET2 homologs) that were consistently upregulated across all ten genotypes in each of the western locations over three years. GmSWEET29 and GmSWEET34 are likely candidates underlying the lower seed protein content of western soybeans. GmSWEET20 (AtSWEET12 homolog) was downregulated in the western locations and may also play a role in lower seed protein content. These findings are valuable for improving soybean agriculture in western growing regions, establishing more strategic and efficient agricultural practices.

## 1. Introduction

The global population is expected to approach 10 billion over the next three decades; the growing demands for crop production to provide food security are driving an urgent need to improve agricultural practices worldwide [1,2]. Yield, quality, and nutrients of crops must carefully balance with climate, land efficiency, regional farming practices, and equipment. Current agricultural practices are inadequate to feed the population, and a global population increase by 25% will add unprecedented stress on global food supplies [1]. Unfavorable climate conditions, increased concentrations of carbon dioxide, water shortages, pests, diseases, and crowded land spaces are among the complications for which humankind must be prepared [3,4]. Understanding soybean’s response to changing climate conditions is important to reduce variability in year-to-year crop yields, promote food supply stability, and food price stability [5,6,7].

Soybean [*Glycine max* (L.) Merr.] is a valuable crop with many agricultural uses, including human food and animal feed. It is a leading source of oil and protein as its seeds have approximately 20% oil, 40% protein, 35% carbohydrates, and 5% ashes [8]. Soybean seeds contain the highest plant protein content of any legume, making seed protein content a key factor in quality standards [9,10]. Soybean quality factors are based on quantitative estimates of seed composition, including protein content, oil content, free fatty acids, chlorophyll, and fatty acid profile.

When considering seed storage macronutrients in soybean, an important balance exists between proteins, lipids, and carbohydrates and it is important to consider expression of genes related to each of these macromolecules. Carbohydrate metabolism is the precursor to protein and oil biosynthesis, making it a key step in downstream seed storage molecule biosynthesis. Seed protein and oil content are complex quantitatively inherited traits that are influenced by genotype, environment, and the interaction between the two [11,12]. A strong indirect phenotypic correlation is evident in the inverse balance of these two storage molecules [13,14].

For nearly two decades a persistent trend has been observed between eastern and western growing regions in Canada; the average soybean seed protein and oil content from western Canada have been consistently low and more variable than the average soybean seed protein and oil content from eastern growing regions of Canada [15]. In 2019, commercially grown soybean from eastern Canada had an average protein content (39.5%) higher than that found in seeds grown in western Canada (36.8%) [16]. In 2020, western-grown soybeans had an average protein content of 37.6% while eastern-grown soybeans had an average protein content of 38.8% [15]. In 2021, average eastern soybean protein content was 40.3%, while the western-grown soybeans had an average protein content of 36.0% [17]. The same trend has been observed between eastern and western Canada for oil. As opposed to the expected inverse relationship seen in seed protein and oil content, the average seed oil content in eastern Canada (22.1%) was higher than the average oil content found in western Canada (21.3%) [13,14,15,18]. Ort et al. [18] compared phenotypic data of ten soybean cultivars from early maturity groups grown in Ottawa (Ontario; east) and Morden (Manitoba; west). Using climate data from both locations the authors correlated environmental factors with differences in development between the two locations. The authors found that the soybeans grown in Morden had longer periods of vegetative growth and less time spent in the flowering and seed development stages, resulting in plants with more leaves compared to their Ontario-grown counterparts [18]. Warmer weather and shorter photoperiod in Ontario lead to a greater rate of phenological development and a positive influence on seed yield [18]. Compounded with the other challenges of growing soybean in western Canada (i.e., drier, cooler, longer photoperiod) the issue with lower protein soybean may result in a loss of interest in soybean production, and subsequently losing a key nitrogen fixing crop from Canadian crop rotation practices.

Photosynthetic tissues are the main source of sugar production, providing carbon energy for the plant. Plants facilitate the transport of sugars across cellular membranes in order to control carbon flux based on supply and demand across various sugar-dependent tissues [19,20]. The Sugars Will Eventually be Exported Transporters (SWEETs) are a family of membrane-bound sugar transporters found on the plasma membranes of tonoplasts and phloem parenchyma cells [21,22,23]. SWEETs function in a concentration-dependent manner to limit loss of sugars via storage in vacuoles [20,24]. SWEET proteins contain a *MtN3-slv* transmembrane domain which is responsible for the maintenance of seed and pollen development, and plant nectar production [25]. Seventeen SWEET genes have been identified in *Arabidopsis thaliana* [19] with homologues found in other plant species, including 52 SWEET genes in the soybean (*G. max*) genome [26], 108 in wheat (*Triticum aestivum)* [27], 21 in rice (*Oryza sativa)* [25], and 32 in wild turnip (*Brassica rapa*) [28].

SWEETs play a role in seed protein and oil accumulation in soybean. GmSWEET10a and 10b alter protein and oil contents and seed size during soybean domestication [29]. *GmST05* (Seed Thickness on Chromosome 5) influences transcription of *gmsweet10a* to regulate seed size, oil, and protein [30]. GmSWEET39 (*Glyma.15G049200*) was recently identified as a seed coat-specific sugar transporter for seed development with a positive correlation to oil content in seeds, suggesting its specialized role in low protein and high oil in soybean seeds [31]. GmSWEET39 was demonstrated to function in both protein and oil improvement under artificial selection in two different paths, postulating its use for oil and protein improvement in soybean. GmSWEET39 was identified as the underlying gene for a major protein QTL on chromosome 15; a CC deletion resulting in a truncated GmSWEET39 was associated with high seed oil and low seed protein content. The CC deletion mutant was extensively used in soybean improvement, particularly in North America. The authors proposed that GmSWEET39 regulates oil and protein accumulation by influencing sugar delivery from the integument of the seed coat to the embryo [31].

Changes in environmental conditions are typically responsible for year-to-year seed composition variation, whereas genetic differences between cultivars are typically responsible for trends spanning several years. Drought and temperature stress play a role in seed protein and oil content [32]. Cober et al. [33] found moderate seed protein heritability, indicating the role environment plays in seed protein accumulation. Combined, environmental variation and soybean genetics are responsible for the differences in seed composition between eastern and western growing regions [15]. There is a limit to the current understanding of oil and storage protein regulation and accumulation in soybean seeds, as well as the determining factors for the allocation of carbon to the production of proteins and oil [34,35]. Here, using RNA-sequencing (RNA-seq) and differential expression (DE) analysis of soybeans grown in eastern and western Canada differentially expressed (DE) GmSWEETs that influence carbon allocation are uncovered.

## 2. Results

### 2.1. Seed Protein, Seed Oil Content, and Yield

Ten genotypes (lines) were planted in four locations across east and west Canada for DE analysis from 2018–2021(Appendix A). Table 1 lists the seed protein and oil content for all 10 genotypes in each location and each year, as a percentage of the total seed at 13% moisture. Over all four locations, line 1 (X5583-1-041-5-5), line 2 (AC Harmony), and line 3 (AAC Halli) had the lowest average seed protein content of 37.5%, 36.7%, and 38.9%, respectively (Table 1). Lines 8 (AAC Springfield), 9 (Jari), and 10 (AC Proteus) had the highest average seed protein content across all lines over the 4 locations, with mean protein contents of 44.6%, 43.0%, and 46.9%, respectively (Table 1). Reciprocally and expectedly, Lines 8, 9, and 10 have the lowest mean oil content (18.5%, 19.0%, and 16.5%, respectively). Lines 1, 2, and 6 (OT13-08) had the highest mean oil content of 22.1%, 22.4%, and 21.6%, respectively (Table 1). Line 1 had the lowest mean protein content across all four locations (37.5%) and the highest yield of 2715 kg ha^−1^, with oil content among the top results (22.1%) (Table 1). Line 10 had the highest average protein content over all four locations (46.9%) while simultaneously having the lowest oil content (16.5%), and the lowest mean yield at 1947 kg ha^−1^ (Table 1).

The mean protein content across all genotypes in the east (Ottawa) was 43.6%, and the mean protein content across all genotypes in the three west locations (Morden, Brandon, Saskatoon) was 40%; a difference of 3.6% (Table 1). Morden, Brandon, and Saskatoon had similar mean protein content (mean of all lines per location over 4 years; 40.6%, 39.9%, and 39.8%, respectively) (Table 1).

The mean oil content of all genotypes in the east was over 20.7%, and the mean oil content of all genotypes in the three west locations together was 20.2% (Table 1). Morden had a higher average oil content across all ten genotypes (21.0%) than Ottawa (20.7%) (Table 1). The average oil content in Brandon over all genotypes was 19.9% and the average oil content in Saskatoon was 19.8%; a difference from the east of 0.8% and 0.9%, respectively (Table 1).

The mean yield in the east over all genotypes was 2665 kg ha^−1^, and the mean yield in the west across all genotypes was 2266 kg ha^−1^, a difference of 399 kg ha^−1^ or a 15% decrease in the west (Table 1). The mean yield in Morden across all genotypes was 2658 kg ha^−1^, only 7 kg ha^−1^ less (<1%) than in Ottawa (Table 1). Brandon had a yield 523 kg ha^−1^ less (approx. 20% lower) than all genotypes grown in Ottawa (Table 1). Saskatoon had the lowest average yield of 2066 kg ha^−1^, a difference of 599 kg ha^−1^ (or 22.5% lower) in comparison to Ottawa (Table 1).

### 2.2. Quality Control

RNA Q30 (99.9% accuracy for base calls) scores of at least 36 were accepted across all samples, indicating high quality base calls for each sample. The average surviving read depth was 29,313,314 reads. The average read survival rate was 98.0%, the lowest survival rate was 90.9%. It is important to note Morden 2021 experienced serious drought stress and was excluded from cross-comparative analysis. GGEbiplot analysis across all year-location datasets determined Morden 2021 to be close to the origin of principal components (PC) 1 and 2 (PC1 and PC2), indicating it shows very little information, hence the data were excluded (Figure 1). Figure 2 shows the PC analysis for samples within each individual DE analysis (colored datapoints); and the cumulative dataset PC analysis per year (grey datapoints).

### 2.3. SWEET Family Analysis

Of the 52 GmSWEET genes (Wm82.a1.0) identified by Patil et al. [26], three were no corresponding ID in the Wm82.a2.0 genome database; GmSWEET3 (Wm82.a1.0 ID *Glyma03g39430*), GmSWEET18 (*Glyma06g21570*), and GmSWEET27 (*Glyma08g48280*).

Across all 88 DE datasets, 34 GmSWEET genes were identified to be DE between eastern and western growing locations (Table 2). In total, 31 known GmSWEET genes were identified to be DE, and three *Arabidopsis* AtSWEET homologs with no corresponding GmSWEET names were identified: *Glyma.09G119100* (AtSWEET17 homolog), *Glyma.19G066300* (AtSWEET17 homolog), *Glyma.06G200200* (AtSWEET17 homolog) (Appendix A; ST1). Of 774 instances of DE SWEET genes across all 88 datasets (4 years, 3 west locations, 10 genotypes) there were 411 instances of upregulation and 363 instances of downregulation in the west (Table 2). The mean upregulated log2FC difference in expression is 3.08 and the median upregulated log2FC is 2.47. The mean downregulated log2FC is −3.38 and the median downregulated log2FC difference in expression is −2.81. GmSWEET12 (*Glyma.05G202600*) in Line 9 in Morden 2020 showed the highest upregulation of any SWEET gene, DE by a log2FC 23.7. GmSWEET12 also had the lowest downregulation of any SWEET gene, as seen in Line 7 in Saskatoon in 2021, DE by a log2FC of −26.9. Figure 3 depicts the ClustalW phylogenetic relationship of the peptide sequences from all 17 AtSWEETs (blue highlight), and all Wm82.a2.0 GmSWEETs (green highlighted GmSWEETs are found DE in our data; yellow highlighted GmSWEETs are known GmSWEETs not found to be DE in our data). In Figure 3, GmSWEET genes highlighted in green with a border have no known GmSWEET identity but are homologs of AtSWEETs. GmSWEETs from all three clades found in soybean were found to be DE between east and west. Clades I and II are hexose transporters and Clade III SWEETs are specific to sucrose transport [36,37].

### 2.4. GmSWEET29 Is Persistently Upregulated in Soybeans Grown in West Locations

ST1 summarizes all the log2FC differences in expression between east (Ottawa) and west (Morden, Brandon, or Saskatoon). DE analysis was done by comparing each genotype grown in Ottawa to each of its western-grown counterparts (identical genotype) within the same year (Appendix A). The most consistently DE SWEET gene across all 88 DE datasets is GmSWEET29 (*Glyma.12G234500*). GmSWEET29 was upregulated in the west in 62 of 88 east vs west DE analyses, spanning nearly all years and locations; GmSWEET29, however, was consistently downregulated across 9 of 10 lines in Saskatoon 2020 and was not DE in Morden 2020 (Table 2, ST1). All ten soybean genotypes exhibited upregulation of GmSWEET29 in the west in two or three years (ST1). Upregulation of GmSWEET29 ranged across all data sets from a log2FC of 0.824 (Brandon Line 9 2021) to a log2FC of 6.42 (Brandon Line 4 2019). The median upregulation of GmSWEET29 between east and west was by a log2FC of 3.45. In Saskatoon 2020 (irrigated), GmSWEET29 was downregulated in 9 of 10 genotypes by a log2FC ranging from −1.57 (Line 9) to −3.14 (Line 8), and an average of −1.97 and a median value of −1.86.

### 2.5. GmSWEET20 and GmSWEET34 Show Some DE between East and West Locations

GmSWEET34 (*Glyma.13G264400*) showed consistent upregulation in 55 of 88 datasets, spanning nearly all western locations in three of four years, with only one instance of downregulation (Line 9, Saskatoon 2020; log2FC −1.66) (Table 2, ST1). The upregulation of GmSWEET34 in the west ranged from a log2FC of 0.651 (Line 6, Saskatoon 2021) to a log2FC of 4.50 (Line 4, Brandon 2019); the median upregulation was by a log2FC 2.47. However, the instances of GmSWEET34 DE are not convincingly persistent, as seen in Brandon 2018 which has only one genotype (Line 6) exhibiting upregulation by a log2FC of 1.87, and no instances of upregulation across all samples in 2020 (ST1).

GmSWEET20 (*Glyma.08G009900*) showed downregulation in 47 of 88 DE datasets, including all locations and all years, however GmSWEET20 was upregulated in 4 genotypes from 2020; Morden Lines 5, 9, and 10, and Saskatoon Line 10 (Table 2, ST1). The downregulation of GmSWEET20 ranged from log2FC −0.881 (Line 5, Brandon 2021) to −8.81 (Line 7, Morden 2019); the median downregulated expression was log2FC −4.77. The DE patterns seen across 2020 samples is not persistent across all genotypes, somewhat hindering the fit of GmSWEET20 into our candidate gene criteria.

### 2.6. Glyma.09G119100, Glyma.19G066300, Glyma.06G200200 Are AtSWEET17 Homologs

Our results uncovered three *G. max* genes not previously identified in the SWEET family in soybean, *Glyma.09G119100*, *Glyma.19G066300*, and *Glyma.06G200200*. *Arabidopsis* homeolog AtSWEET17 was identified as the top TAIR10 result for these three *G. max* genes (ST1). *Glyma.06G200200* is uncharacterized in soybean but the top BLASTP hit identifies bidirectional sugar transporter SWEET17 (AtSWEET17) in *A. thaliana* as the most closely related sequence (Figure 3, ST1). *Glyma.09G119100* is identified as a predicted bidirectional sugar transporter SWEET17-like protein in *G. max* (ST1). *Glyma.19G066300* is also uncharacterized in soybean and its top BLASTP hit identified Brassinosteroid-regulated protein (BRU1) in *G. max* as the most closely related protein (ST1). These three AtSWEET17 homologs were DE only in 2020 and 2021 (ST1). Line 2 in Saskatoon 2020 had upregulated expression of *Glyma.09G119100* and Lines 3 and 8 in Saskatoon 2021 had downregulated expression of *Glyma.09G119100*. Line 5 in Saskatoon 2021 had downregulated expression of *Glyma.06G200200*. In 2020, Morden Line 7 had downregulated expression of *Glyma.19G066300* and Line 10 upregulated expression of *Glyma.19G066300*. Line 10 in Saskatoon 2020 had upregulated expression of *Glyma.19G066300*. Line 4 in Brandon 2021 had upregulated expression of *Glyma.19G066300,* while Line 10 in Saskatoon 2021 had downregulated expression of *Glyma.19G066300*.

### 2.7. Top DE SWEET Genes between East and West

The log2 fold change (log2FC) difference in expression across each individual genotype (Lines 1–10) between Ottawa (east) and each western location (Morden, Brandon, Saskatoon) across all 4 years is summarized in ST1, including the full datasheet with precise log2FC values. Additionally, the soybase.org [38] genome annotations (BLASTP, TAIR10, gene ontology (SoyBase and PANTHER), protein family descriptions (PFAM), and eukaryotic orthologous groups (KOG)) are also listed in ST1.

### 2.8. Weather Report

Figure 4 shows the average temperature (°C) for each location from 2018–2021. The 7-day moving average for each location is indicated by solid lines and the individual datapoints represent the daily highs and lows. Appendix A (SF2) shows the relative humidity (%) of the experimental farms every day from 2018–2021. The 7-day moving average for each location is indicated by solid lines and individual datapoints represent the daily values. Appendix A (SF3) shows the daily precipitation totals (mm) at each experimental farm during growing seasons (mid-April to mid-November). For each of Figure 4, SF2, and SF3 the east (Ottawa) data are in green, and west (Morden, Brandon, Saskatoon) data are in shades of blue. The average temperature during the growing seasons across 2018–2021 in Morden was 11.0°C, Brandon was 9.2°C, Saskatoon was 8.9°C, and Ottawa was 13.6°C. The average relative humidity during the growing seasons across 2018–2021 in Morden was 62.7%, Brandon was 66.7%, Saskatoon was 63.7%, and 68.2% in Ottawa. Cumulative rainfall from mid-April to mid-November in Ottawa was 548 mm in 2018, 544 mm in 2019, 438 mm in 2020, and 447 mm in 2021, totaling 1977 mm over 4 years. Seasonal rainfall in Morden was 306 mm in 2018, 476 mm in 2019, 214 mm in 2020, and 329 mm in 2021, totaling 1325 mm over the 4 years. In Brandon, seasonal rainfall was 220 mm in 2018, 540 mm in 2019, 423 mm in 2020, and 394 mm in 2021, totaling 1576 mm over the 4 years. Saskatoon had the lowest seasonal rainfall with 165 mm in 2018, 232 mm in 2019, 259 mm in 2020, and 152 mm in 2021, a total of 807 mm over the 4 years.

## 3. Discussion

### 3.1. GmSWEET29 Is an AtSWEET2 Homolog and Is Upregulated in Soybeans Grown in the West

GmSWEET29 (*Glyma.12G234500*) was upregulated in western-grown soybeans in 62 of the 88 east vs west DE analyses, evident in all west locations over three years (ST1). To the best of our knowledge there are no reports of GmSWEET29 implicated for a role in seed protein. Patil et al. [26] describes AtSWEET2 as a homolog of GmSWEET29 (*Glyma12g36300*). Our results corroborate these findings, as the TAIR10 result for *Glyma.12G234500* is AtSWEET2; Nodulin MtN3 family protein (ST1) and as seen by the phylogenetic clustering of GmSWEET29 and AtSWEET2 in clade I (Figure 3).

AtSWEET2 belongs to clade I of the SWEET family of sugar transporters, alongside plasma membrane hexose transporter AtSWEET1 [19,22,36,39] (Figure 3). AtSWEET2 is a plasma membrane hexose transporter, identified to play a role in accumulation of glucose in plants [24]. Similarly, the rice homolog OsSWEET2b was found to localize to the tonoplast and also played a role in glucose transport mediation in lipid vesicles, corroborating other findings that SWEET2 homologs function as hexose transporters in the tonoplast [22]. Glucose trafficking was increased in lipid vesicles with OsSWEET2b when compared to controls, and glucose trafficking activity was eliminated when mutations to block the sugar transport channel of OsSWEET2b were introduced [22,40]. Crystal structure analysis of OsSWEET2b identified an asymmetrical pair of triple-helix bundles connected by an inversion linker transmembrane helix to form the pathway for sugar translocation [22].

Guo et al. [41] identified AtSWEET2 accompanied by vacuolar transporters AtSWEET16 and AtSWEET17 as the most highly expressed AtSWEETs in *Arabidopsis* root tissue [41]. AtSWEET2 fused with GUS and GFP reporter proteins was identified to function in the root tip and root cap tonoplasts such as AtSWEET16 and 17, and were particularly localized to the cortex and epidermis [24]. It was suggested that AtSWEET2 plays a role in glucose sequestration of the root vacuoles by moderating carbon efflux and preventing loss of sugar within roots [24]. This is important because a significant portion of sugars are lost from root tissue into the rhizosphere, which serves as a carbon source for rhizobacteria as well as for soil-borne pathogenic microbes [24,42]. Stable isotope analysis using loss-of-function *atsweet2* mutants showed an increase in the amount of glucose lost from roots to the rhizosphere [24]. Conversely, overexpression of AtSWEET2 limited carbon efflux from roots by accumulating sugars in vacuoles, which notably contributed to increased resistance to the soil-borne oomycete, *Pythium irregulare* [24].

It is important to acknowledge the majority of what is described today about AtSWEET2 is focused primarily on root tissue or expressed in yeast lipid vesicles, and our data used leaf tissue. While its role in root sugar accumulation and its relationship to the rhizosphere has been established, our data presents that the AtSWEET2 soybean homolog GmSWEET29 is obviously DE between soybeans grown in east and west locations of Canada. This suggests that GmSWEET29 may play a role in mediating efflux of sugar in leaf vacuoles. The Bio-Analytic Resource for Plant Biology (BAR; http://bar.utoronto.ca/; accessed on 28 August 2022) ePlant Soybean eFP Viewer indicates expression of GmSWEET29 (*Glyma.12G234500*) is highest in the root hairs and flowers and moderately expressed in the leaves. Expression data for GmSWEET34 (*Glyma.13G264400*) indicated very low expression in the roots and root hairs and high expression in the leaves [43,44].

From our data, there is an increase in expression of hexose transporter GmSWEET29 in soybeans grown in the west. Based on what is known of SWEET2 homologs, GmSWEET29 may be present on tonoplasts and play a role in sugar accumulation in vacuoles.

### 3.2. GmSWEET34 Is a GmSWEET29 Paralog and a AtSWEET2 Homolog

GmSWEET34 (Glyma.13G264400) was consistently upregulated in 55 of 88 datasets, with only one instance of downregulation (ST1). At least 9 genotypes showed upregulation of GmSWEET34 across 5 of the 9 location-year metadata: Morden-2018, Morden-2019, Brandon-2019, Saskatoon-2019, and Brandon-2021. However, in Brandon-2018 only Line 6 showed DE from Line 6 in Ottawa, and the remaining 9 lines did not have any significant DE of SWEET genes. In Saskatoon-2021, Lines 2, 4, 6, 7, and 8 had upregulated expression of GmSWEET34 compared to their Ottawa-grown counterparts, while Lines 3, 5, 9, and 10 did not yield DE for GmSWEET34.

GmSWEET34 is not well characterized in *G. max* and the top BLASTP hit is ruptured pollen grain in *Medicago truncatula* (ST1). Interestingly, like GmSWEET29, GmSWEET34 is an AtSWEET2 (Nodulin MtN3 family protein) homolog as per our TAIR10 results and as previously reported by Patil et al. [26]. Figure 3 shows GmSWEET29, GmSWEET34, and AtSWEET2 cluster closely together in clade I (hexose transporters). This suggests that GmSWEET34 also plays a role in sugar accumulation in vacuoles and limiting efflux.

The genetic evolution of modern soybean includes two whole genome duplications [45,46,47,48]. Soybean has notably more SWEET genes (52) than model plants such as *Arabidopsis* (17) and rice (21) [19,25,26]. However, Patil et al. [26] identified 21 GmSWEET paralogous pairs in *G. max*. By assessing synonymous (Ks) and non-synonymous (Ka) substitution mutation rates, and the ratio of the two (Ks/Ka), Patil et al. [26] elucidated GmSWEET gene pairs with highly similar coding sequence alignments and estimated the approximate time in history at which the duplication event took place. GmSWEET29 and GmSWEET34 had a Ka/Ks of 0.233, where a Ka/Ks < 1 implies that the genes underwent stabilizing (or purifying) selection. GmSWEET29 and GmSWEET34 were estimated to have been duplicated approximately 8.05 million years ago [26]. This is significant to our findings as the upregulation of GmSWEET29 and GmSWEET34, both AtSWEET2 homologs, are highly prevalent in our data. Both genes were significantly upregulated across the three western locations across three years, indicating that soybeans grown in the west have an increasing expression of GmSWEET29 and GmSWEET34.

Expression of GmSWEET29 and GmSWEET34 was generally upregulated in 2018, 2019, and 2021 (ST1). Thus, it is interesting that GmSWEET29 is downregulated in nine samples in Saskatoon 2020 and not DE in Morden 2020; GmSWEET34 is downregulated in one sample in Saskatoon 2020 and is not DE in Morden 2020 (ST1). While expression of GmSWEET29 and GmSWEET34 in 2020 stand out from 2018, 2019, and 2021, this lack of DE seen in Morden 2020 and downregulation seen in Saskatoon 2020 indicate that GmSWEET29 and GmSWEET34 expression patterns changed in a similar manner year over year (i.e., no longer upregulated). GmSWEET29 and GmSWEET34 are both clade I hexose transporter SWEETs, thereby indicating that the expression patterns of hexose transporters is an important feature underlying the differences between eastern and western grown soybeans.

### 3.3. GmSWEET20, a AtSWEET12 Homolog, Is Downregulated in the West

GmSWEET20 (*Glyma.08G009900*) was consistently downregulated in 47 of 88 DE datasets, spanning all western locations in three years (ST1). *Glyma.08G009900* is not well characterized in *G. max*, and the top BLASTP hit identified the bidirectional sugar transporter SWEET12 from *Phaseolus vulgaris* as the most closely related protein (ST1). AtSWEET12 is the top *Arabidopsis* homolog as determined by TAIR10, and Figure 3 shows GmSWEET20 is a clade II SWEET hexose transporter and clusters most closely with AtSWEET12 (ST1, Figure 3). All genotypes (lines 1–10) in Saskatoon-2019 significantly downregulated their expression of GmSWEET20 in comparison to their eastern counterparts (ST1). Saskatoon-2020, Brandon-2021, and Saskatoon-2021 had three or four lines that downregulated GmSWEET20 expression. Morden-2018, Brandon-2018, Morden-2019, and Brandon-2019 had between 5 and 8 lines that downregulated the expression of GmSWEET20. Morden-2020 had one line that downregulated GmSWEET20 expression.

Patil et al. [26] identified GmSWEET20 and GmSWEET12 as sister paralogs, with a Ks/Ka of 0.300 and an estimated duplication event 8.68 million years ago. GmSWEET12 is DE in 26 of 88 line-location-year datasets: 19 instances of downregulation in the west, and 7 instances of upregulation in the west (Table 1, ST1). GmSWEET20 and GmSWEET12 followed similar patterns across all four years of our study (ST1). In the majority of instances these genes are downregulated in the west; however in 2020, the same change in expression (upregulation) in the same lines (Morden Line 5, Morden Line 9, Morden Line 10, and Saskatoon Line 10) was observed. Because GmSWEET20 and GmSWEET12 are sister paralogs, it is not surprising that their expression patterns across different genotypes were similar. AtSWEET12 was identified along with AtSWEET11 as a clade II hexose sugar transporter, localized to the plasma membrane of parenchyma cells of the phloem and are key players in phloem loading [23]. Because cold treatment is known to induce sugar accumulation, one study analyzed the behavior of different *Arabidopsis* mutant lines after 1 week at 4 °C. The double mutant for *atsweet11-12* was found to exhibit a greater cold tolerance than the wild type and both single mutants; this study confirmed the role of clade II hexose transporters AtSWEET11 and AtSWEET12 in the regulation of sugar transport [49]. A decrease in expression of phloem-loading parenchyma membrane sugar transporters suggests plants in the west may have significantly reduced the loading of sugars into phloem for carbon transportation to various carbon sinks.

AtSWEET12, AtSWEET11, and AtSWEET15 together play a distinct role in seed filling [50]. An *atsweet11/12/15* triple mutant contained the same number of seeds but had a reduced dry seed weight of ~43% in comparison to the control plants. The AtSWEET11 and AtSWEET12 genes were found to be expressed in both leaves and seeds, hence, the triple mutant had a lack of carbohydrates in the leaves which directly correlated to a reduced supply to be exported, negatively impacting embryo development and seed filling [51]. A mutation found in *atsweet11/12/15* genes in *Arabidopsis* impaired sucrose delivery from seed coat and endosperm to the embryo which resulted in many defects to the seed [50]. In a similar manner, a knockout mutation in the rice genes *ossweet11* and *15* resulted in complete loss of endosperm development [52,53]. In soybean, a GmSWEET15b knockout led to a high rate of seed abortion [54]. These studies suggest that downregulation or knockout of SWEET11/12/15 has a significant impact on seeds, including their viability and development. In our study persistent DE of AtSWEET12 homolog GmSWEET20 and sister paralog GmSWEET12 in soybeans grown in the west of Canada was identified. The DE of GmSWEET20 and GmSWEET12 observed between east and west locations is valuable for understanding the cause of the lower seed protein phenotype seen in western Canada.

### 3.4. Glyma.06G200200, Glyma.09G119100, and Glyma.19G066300 May Be SWEET-like Proteins

Three AtSWEET17 homologs were identified in our data, with no known GmSWEET identity, *Glyma.06G200200*, *Glyma.09G119100*, and *Glyma.19G066300* (ST1). AtSWEET17 is a clade III (sucrose transporter) SWEET (Figure 3) [26,37]. The three putatively new GmSWEET genes clustered in clade III with AtSWEET16 and 17, as well as GmSWEET19, 28, 40, 50, 52 (Figure 3). AtSWEET17 was described as a facilitative transporter of fructose on root and leaf tonoplasts along with AtSWEET16 [41]. Mutations affecting AtSWEET17 expression have led to fructose accumulation in leaves confirming its role in exporting fructose from leaf vacuoles. However, interestingly, AtSWEET17 was not highly expressed in leaves which indicated that it functioned in other organs of the plant. qRT-PCR studies revealed that AtSWEET17 was highly expressed in plant roots and the expression of AtSWEET17 was considerably higher during lateral root growth under drought conditions, and vice versa [55]. Another study identified a possible correlation between overexpressing EjSWEET17 and the amount of fructose and sorbitol contents in loquat (*Eriobotrya japonica*) fruit indicating a relationship with mature fruit sugar content [56]. Next steps for this research should include verifying the function of *Glyma.06G200200*, *Glyma.09G119100*, and *Glyma.19G066300* as sugar transporters.

### 3.5. GmSWEET39 Is Not DE between Eastern and Western-Grown Soybeans in Canada

GmSWEET39 (*Glyma.15G049200*), an AtSWEET15 homolog, was not DE between eastern- and western-grown soybeans in our data. GmSWEET39 underlies an important soybean protein QTL, in which a 2 bp CC deletion was found to be associated with high seed oil and low seed protein content [31]. GmSWEET39 is preferentially expressed in seed coat and was identified to have pleotropic effects on seed protein and oil [31]. This preferential expression being localized to the seed coat may explain the lack of DE GmSWEET39 in our work. Alternatively, this QTL may not have significant importance in Canadian short-season soybean [57]. Patil et al. [26] identifies GmSWEET39 and GmSWEET24 as sister paralogs; GmSWEET24 is not found to be DE in our data.

### 3.6. SWEET Genes Have Been Shown to Influence Yield

SWEETs have been recently identified to play a role in yield and biomass accumulation [30,58]. Expression analysis of SWEETs in both maize and foxtail millet identified a higher sucrose transport capacity in maize and that was critical for yield and biomass accumulation [58]. In both rice (*O. sativa* L.) and maize *(Z. mays*), hexose transporter SWEET4 mutations to OsSWEET4 and ZmSWEET4c respectively, lead to defective seed filling in both species, suggesting SWEET4 is important for increased import of sugars into endosperms [59]. Low concentration of sucrose in the developing embryo was the result of decreased sugar transport in both OsSWEET11 rice knockout mutants and GmSWEET15 soybean knockout mutants [52,54]. A recent study identified significant differences between the physiology of soybeans grown in western Canada compared to the same genotype grown in eastern Canada; western soybeans were taller, yields were lower, seeds were smaller, and the amount of fixed nitrogen protein was lower [60]. With this information it is important to consider that the DE GmSWEETs identified in this study likely play a role influencing the difference in seed yield between east and west Canada.

### 3.7. Abiotic and Biotic Considerations

The environmental conditions in Ottawa include more precipitation, higher humidity, and warmer temperatures, favoring soybean growth over the more arid environments in the west (Figure 4, SF2, SF3). These conditions innately provide more optimal conditions for successful soybean growth, as soybean ancestry is East Asian, and Canadian geography is limited to a slim range of northern climate conditions. Taking into consideration the genotypes that are more tolerant to inclement environments is essential to optimizing soybean agriculture in western growing regions. Further, biotic differences between east and west growing locations are a likely factor influencing soybean growth. The microbial rhizosphere has been established to add intense competition for carbon and sugar reuptake in maize in comparison to plants grown in sterile conditions [61]. A devastating pest, Soybean cyst nematode (*Heterodera glycines*), was recently discovered in Manitoba, only a few years after its presence in eastern Ontario was discovered [62,63] which may further complicate the exchange of carbon in the rhizosphere. It would be of interest to investigate the soil microbiomes of east and west soybean growing regions to gain a greater appreciation for the biotic factors at play.

## 4. Materials and Methods

### 4.1. G. max Lines

Ten maturity group 00 [64,65] *G. max* genotypes ranging from low to high in seed protein content were selected for this study; Line 1 has the lowest average seed protein content and Line 10 has the highest seed protein content. Line 1 is X5583-1-041-5-5, Line 2 is AC Harmony [66], Line 3 is AAC Halli, Line 4 is 90A01 [67], Line 5 is Maple Amber, Line 6 is OT13-08, Line 7 is OT14-03, Line 8 is AAC Springfield, Line 9 is Jari, Line 10 is AC Proteus [68]; all developed at the Ottawa Research and Development Centre by Agriculture and Agri-Food Canada (AAFC).

### 4.2. Planting and Growth

Four geographical locations in Canada were chosen for this study: Ottawa (Ontario), Morden (Manitoba), Brandon (Manitoba), and Saskatoon (Saskatchewan) (SF1A). Ottawa (latitude 45.39, longitude −75.72) represents the eastern growing region of Canada, and Morden (49.18, −98.08), Brandon (49.86, −99.98), and Saskatoon (52.15, −106.57) represent the western soybean growing regions of Canada. All ten lines were planted with four replicates in each location over four years from 2018 to 2021, with the exception of Saskatoon in 2018 and Brandon in 2020. Each trial was carried out using 4 × 5 rectangular lattices, including four replicates per line per site. Each plot was planted at a rate of 50 seeds per m^2^. Each site followed best management practices for optimal fertility, planting date, and control of weeds. The Saskatoon site was irrigated in 2020 and 2021.

### 4.3. Sampling and Seed Content Measurements

Sampling of each line was done from three different replicates at each location. Young trifoliate leaves were collected from otherwise-heathy looking plants during R5 [69]. Leaf tissue was rolled into a sterile 1.5 mL microfuge tube and flash frozen in liquid nitrogen in the field. Samples were stored at −80 °C in Ottawa until the samples from Morden, Brandon, and Saskatoon arrived by expedited shipping on dry ice 24 h later, and then stored at −80°C. A total of 100 seeds per plot were analyzed for seed protein and oil content using an Infratec 1241 Grain Analyzer (FOSS North America, Eden Prairie, Minnesota). Due to the extensive size of this study and to avoid significant plot disruption, leaf tissue was selected as the tissue of choice.

### 4.4. RNA Extraction and cDNA Library Preparation

Samples were crushed directly in their respective microfuge tubes using a sterile nuclease-free mortar and liquid nitrogen. The RNA extraction was carried out using the SPLIT total mRNA extraction kit (Lexogen, Vienna, Austria); however the RNA samples from 2020 were extracted by the Génome Québec, McGill University (Montréal, QC, Canada). RNA quality was assessed using a Nanodrop™ 2000 Spectrophotometer (Thermo Fisher Scientific, Waltham, MA, USA), TapeSation 4200 RNA ScreenTape (Agilent, Santa Clara, CA, USA) and 2100 Bioanalyzer (Agilent, Santa Clara, CA, USA) at Génome Québec (Montréal, QC, Canada). Line 1 samples from 2021 did not pass RNA QC and were not included in DE analysis. The E0 mix of the Spike-in RNA variants (SIRVs) (Lexogen, Vienna, Austria) were integrated with the RNA samples as controls to monitor and compare parameters including sensitivity and quantification. Reverse transcription was carried out by the Canadian Centre for Computational Genomics (Montréal, QC, Canada).

### 4.5. RNA-Sequencing

Paired-end sequence reads were assessed for RNA-seq quality control (QC) using dupRadar [70] (Bioconductor, R) to analyze duplication rate across reads. Sample-level read normalization was done using the rlog function from edgeR [71] prior to exploratory data analysis (EDA) using R. Preseq [72] was used to estimate sequencing depth while simultaneously scanning libraries for complexity.

For comprehensive RNA-seq read data evaluation, RSeQC [73,74] was used. Subread featureCounts [75] was used to map reads to the Wm82.a2.v1 reference genome, including genes, exons, promoters, gene bodies, and chromosomal locations. GenPipes is the main in-house framework of the Canadian Centre for Computational Genomics used to perform major processing steps [76]. Read alignment and mapping was done using STAR [77]. Adapter sequences, primers, polyA tails etc., were removed using Cutadapt [78]. FastQC [79] was used to gain a comprehensive view of the sequence read counts, sequence quality histograms, per-sequence and per-base quality scores, sequence length distribution, and overrepresented sequences. HTseq [80] was used to quantify expression on the gene level.

### 4.6. Differential Expression (DE) Analysis

DESeq2 [81] was used for DE analysis. For all DE analyses, the samples in Ottawa were used as the control expression, and the western samples were used as experimental expression (SF1B); therefore, when discussing the log2 fold change (log2FC) difference in expression, up- and downregulation of genes is describing the expression in the west relative to Ottawa. In order to find the most likely candidate underlying the differences in protein content across eastern and western growing regions, the gene(s) were most consistently DE sought out. An ideal candidate gene is consistently DE (1) across all years, (2) in all 3 western locations, and (3) in the same DE orientation (up- or downregulated).

Accession numbers

Accession number type ID: Wm82.a2.0 and NCBI gene IDs

*Glyma.04G198400*  100790580

*Glyma.04G198600*  NA

*Glyma.04G238100*  100775822

*Glyma.04G241400*  100785977

*Glyma.05G202700*  100810946

*Glyma.05G202600*  100787235

*Glyma.06G122200*  100800304

*Glyma.06G125800*  100813119

*Glyma.06G166800*  100810426

*Glyma.06G166900*  100127405

*Glyma.06G167000*  100810962

*Glyma.06G200800*  NA

*Glyma.08G009900*  100801430

*Glyma.08G010000*  100805676

*Glyma.08G025100*  100802303

*Glyma.08G360400*  100799457

*Glyma.09G043200*  100808537

*Glyma.12G234500*  100776187

*Glyma.13G041300*  100785477

*Glyma.13G037900*  100777281

*Glyma.13G169700*  102666983

*Glyma.13G264400*  100305842

*Glyma.14G120300*  100799083

*Glyma.14G159900*  100782301

*Glyma.14G160100*  100785324

*Glyma.15G149800*  100776607

*Glyma.15G210400*  100798392

*Glyma.15G211800*  100790469

*Glyma.18G301200*  100779335

*Glyma.19G009900*  100811287

*Glyma.20G082700*  100782814

*Glyma.09G119100*  NA

*Glyma.19G066300*  100816986

*Glyma.06G200200*  NA

## 5. Conclusions

Using 88 individual DE analyses the DE of GmSWEET genes in ten soybean genotypes grown in east and west locations of Canada were investigated. GmSWEET genes are DE between east and west grown soybeans, indicating that there may be differences in carbon allocation. Western-grown soybeans increase their expression of GmSWEET29 and GmSWEET34 (AtSWEET2 homologs). GmSWEET20 and GmSWEET12 (AtSWEET12 homologs) are downregulated in soybeans grown in the west. These findings are valuable for future research to increase soybean protein in western Canada. It would be of scientific interest to genetically modify expression of GmSWEET29 and GmSWEET34 in soybean in such a way that the sugar transport channel is disrupted to determine whether protein content, yield, or both would be affected.

## Figures and Tables

**Figure 1 plants-11-02337-f001:**
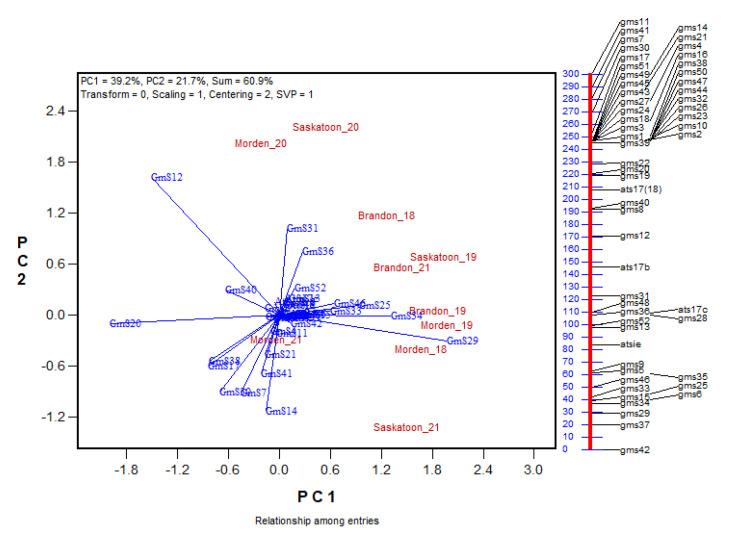
GGEbiplot of all year-location datasets to assess congruency of experimental data sets.

**Figure 2 plants-11-02337-f002:**
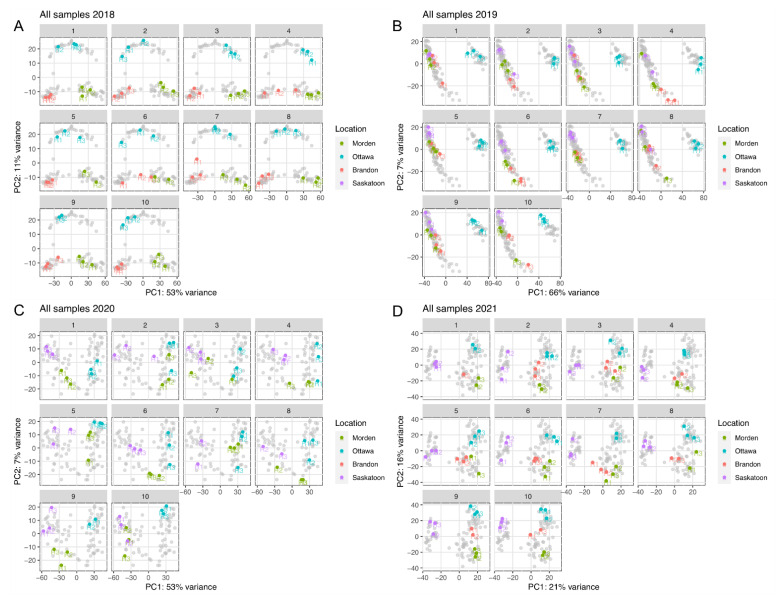
Principal component (PC) analysis of expression data variation for each line across all locations from (**A**) 2018, (**B**) 2019, (**C**) 2020, and (**D**) 2021. Colorful datapoints represent samples in each individual DE dataset; grey datapoints represent all datapoints across all ten lines in a year.

**Figure 3 plants-11-02337-f003:**
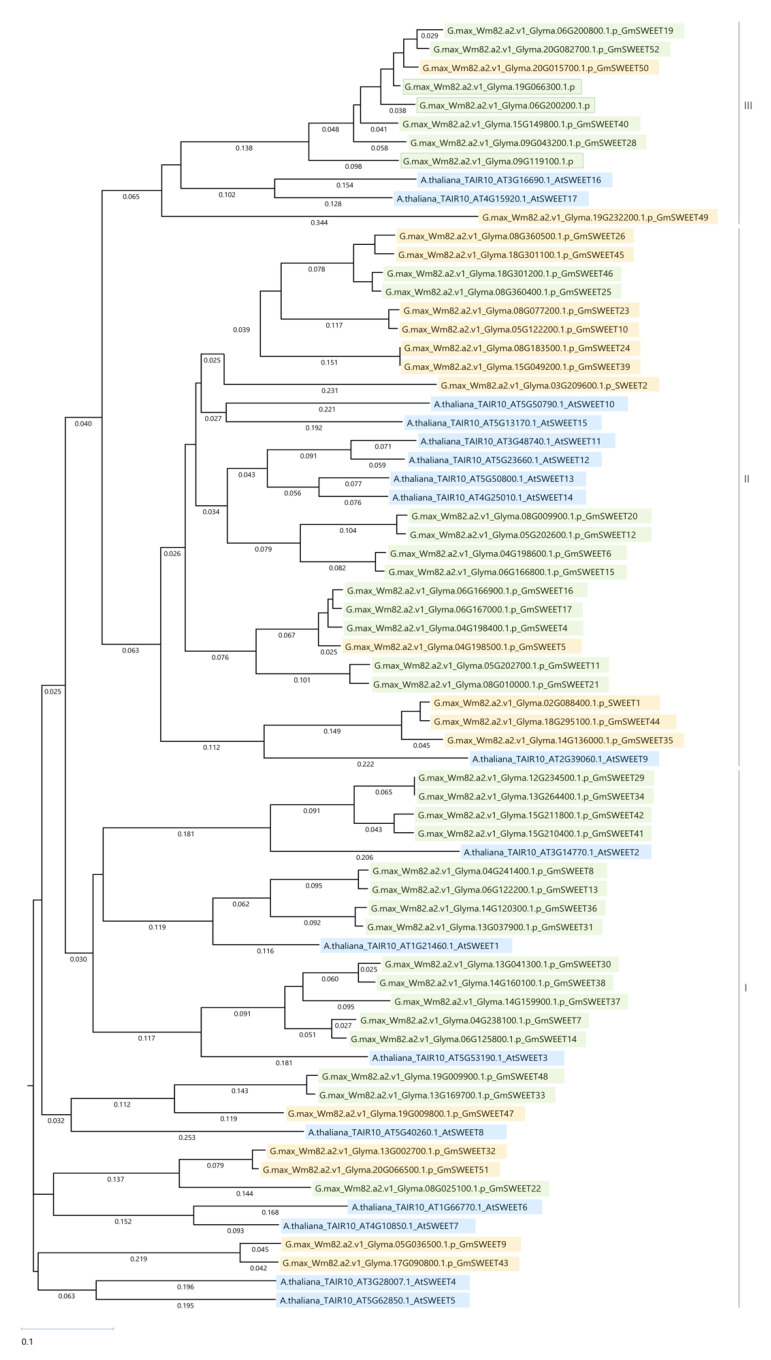
Peptide sequence phylogenetic tree summarizing the relationships between all *A. thaliana* AtSWEETs (blue highlighting), and *G. max* GmSWEETs (genes with green highlighting are found to be DE in our data, genes with yellow highlighting are not). *G. max* genes not previously identified to be a GmSWEET are highlighted in green with a green border (clade III). SWEET clades are indicated by roman numerals and vertical lines spanning each group; clades I and II are subgroups of hexose transporters, clade III are sucrose transporters. MegAlign Pro^®^ Version 17.3 (DNASTAR Lasergene, Madison, Wisconsin) multiple sequence ClustalW alignment and Neighbour-joining methods with 1000 bootstrap replicates were used to construct the tree.

**Figure 4 plants-11-02337-f004:**
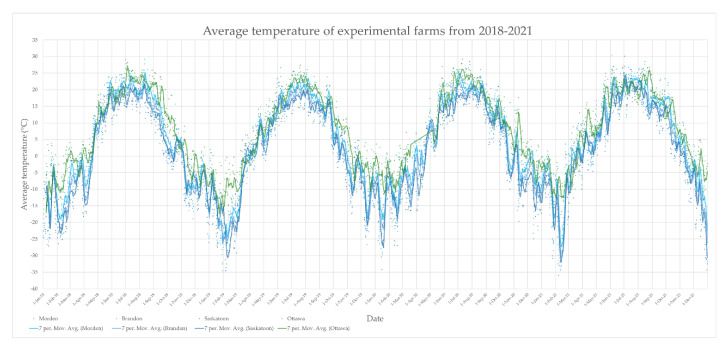
Average temperature (°C) of experimental farms from 2018–2021. East (Ottawa) data is represented in green, west (Morden, Brandon, Saskatoon) data are represented in shades of blue. Daily temperature data are represented by individual data points and the 7-day moving average is represented by solid lines.

**Table 1 plants-11-02337-t001:** Seed protein, oil, and yield for all lines in each location from 2018–2021. Values for protein and oil are listed in average % of total seed. ND = No data. Please note, “pro” indicates seed protein content, “oil” indicates seed oil content, and “yield” indicates the average yield (kg ha^−1^) (raw data collected at the individual line-location-year level, not shown). Mean values for protein, oil, and yield are given as mean per line per location over 2018–2021, mean of all lines per location (1-10), the mean of all lines across all three western locations (west; Morden, Brandon, Saskatoon), and the mean per line over four locations (east + west; Ottawa, Morden, Brandon, Saskatoon).

		Ottawa			Morden			Brandon			Saskatoon			West			East + West		
Line(s)	Genotype	Pro	Oil	Yield	Pro	Oil	Yield	Pro	Oil	Yield	Pro	Oil	Yield	Pro	Oil	Yield	Pro	Oil	Yield
1	X5583-1-041-5-5	40.4	22.8	2912.8	27.5	17.1	2214.3	35.6	21.9	2572.1	36.5	20.9	2447.2	33.2	20.0	2411.2	37.5	22.1	2715.2
2	AC Harmony	38.9	23.4	2589.7	27.2	17.4	1677.6	34.8	21.8	1866.0	36.2	21.5	2229.8	32.7	20.2	1924.5	36.7	22.4	2256.1
3	AAC Halli	41.6	21.7	2745.7	28.8	16.4	2110.9	38.1	20.5	2395.8	37.2	20.8	2253.3	34.7	19.2	2253.3	38.9	21.2	2544.8
4	90A01	42.3	21.4	2523.5	29.8	16.3	1803.5	39.4	20.8	2001.2	38.8	20.6	2244.0	36.0	19.2	2016.2	40.1	21.1	2306.2
5	Maple Amber	42.8	21.8	2516.9	29.8	16.5	1859.1	39.2	20.7	2222.3	38.5	20.5	1893.6	35.8	19.2	1991.7	40.1	21.2	2267.5
6	OT13-08	43.8	21.7	2934.5	30.8	16.6	2265.5	40.3	21.4	2525.1	39.5	21.2	2214.6	36.9	19.7	2335.1	41.3	21.6	2659.6
7	OT14-03	43.7	20.2	2742.1	31.1	15.4	2110.7	41.3	18.8	2090.4	40.0	19.5	2074.6	37.4	17.9	2091.9	41.6	19.8	2427.2
8	AAC Springfield	47.3	18.3	2365.2	32.6	14.5	1801.7	42.5	18.4	2000.1	44.4	18.0	1941.1	39.8	17.0	1914.3	44.6	18.5	2173.7
9	Jari	46.4	19.2	3054.2	31.9	14.7	2267.8	41.3	18.7	2087.6	41.3	18.5	1894.8	38.1	17.3	2083.4	43.0	19.0	2509.3
10	AC Proteus	49.4	16.6	2271.7	34.9	12.7	1824.2	45.9	16.1	1658.2	45.4	16.5	1475.0	42.1	15.1	1652.5	46.9	16.5	1947.0
1–10		43.6	20.7	2665.6	40.6	21.0	2658.0	39.9	19.9	2141.9	39.8	19.8	2066.8	40.0	20.2	2266.7	41.1	20.3	2380.7

**Table 2 plants-11-02337-t002:** All GmSWEET genes DE (adjusted *p*-value ≤ 0.05) between eastern and western regions in at least 1 line in at least 1 location in any year(s). Of 88 DE datasets, the total number of data sets in which each GmSWEET gene is DE is listed under “total”. The number of datasets in which the gene is upregulated in the west is listed under “up”, and the number of datasets where each gene is found to be downregulated in the west is listed under “down”.

Name	Gene ID (Wm82.a2.0)	Up	Down	Total
GmSWEET4	*Glyma.04G198400*	4	1	5
GmSWEET6	*Glyma.04G198600*	32	3	35
GmSWEET7	*Glyma.04G238100*	0	27	27
GmSWEET8	*Glyma.04G241400*	7	2	9
GmSWEET11	*Glyma.05G202700*	6	16	22
GmSWEET12	*Glyma.05G202600*	7	19	26
GmSWEET13	*Glyma.06G122200*	10	5	15
GmSWEET14	*Glyma.06G125800*	5	31	36
GmSWEET15	*Glyma.06G166800*	26	3	29
GmSWEET16	*Glyma.06G166900*	1	0	1
GmSWEET17	*Glyma.06G167000*	2	28	30
GmSWEET19	*Glyma.06G200800*	1	1	2
GmSWEET20	*Glyma.08G009900*	4	47	51
GmSWEET21	*Glyma.08G010000*	6	25	31
GmSWEET22	*Glyma.08G025100*	2	6	8
GmSWEET25	*Glyma.08G360400*	25	0	25
GmSWEET28	*Glyma.09G043200*	24	12	36
GmSWEET29	*Glyma.12G234500*	62	9	71
GmSWEET30	*Glyma.13G041300*	5	34	39
GmSWEET31	*Glyma.13G037900*	19	4	23
GmSWEET33	*Glyma.13G169700*	23	0	23
GmSWEET34	*Glyma.13G264400*	55	1	56
GmSWEET36	*Glyma.14G120300*	14	1	15
GmSWEET37	*Glyma.14G159900*	4	0	4
GmSWEET38	*Glyma.14G160100*	2	38	40
GmSWEET40	*Glyma.15G149800*	8	26	34
GmSWEET41	*Glyma.15G210400*	7	13	20
GmSWEET42	*Glyma.15G211800*	17	5	22
GmSWEET46	*Glyma.18G301200*	17	0	17
GmSWEET48	*Glyma.19G009900*	3	0	3
GmSWEET52	*Glyma.20G082700*	9	1	10
AtSWEET17	*Glyma.09G119100*	1	2	3
AtSWEET17	*Glyma.19G066300*	3	2	5
AtSWEET17	*Glyma.06G200200*	0	1	1
Total		774	411	363

## Data Availability

Not applicable.

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
