# Peer review of "GmSWEET29 and Paralog GmSWEET34 Are Differentially Expressed between Soybeans Grown in Eastern and Western Canada"

_plants, 2022, doi:10.3390/plants11182337_

Round 1
Reviewer 1 Report
The manuscript is poorly written and is full of grammatical mistakes. With such a poor english, it is impossible to judge the merit of the manuscript. A few suggestions/observations are presented below:
Abstract: Introductory part in the abstract is unwarranted. Delete such lines. Focus more on the work done and the results.
Introduction: Line 38-51: Only 1-2 sentence seems enough. Rest can be deleted.
Lines 23, 26, 27-30, 133-135, 594-595, amd many more: Sentence meaning is not clear.
This manuscript needs rewriting for each section.
Author Response
We thank the Reviewer#1 for taking the time to review our manuscript. Their feedback is much appreciated and has been considered upon editing. The Reviewer suggested careful consideration of the grammar used throughout the manuscript. The manuscript has been diligently combed for accuracy and context.
The abstract has been edited for clarity to the reader and the introductory sentence has been removed.
in particular, The Reviewer suggests reducing lines 38-51 to only 1-2 sentences. A number of sentences have been removed, however we feel it is important to stress the need for strategic and improved agricultural practices due to the pressing nature of food shortages, inflation and rising population. Some readers may not be agricultural plant scientists and this information may be impactful to understanding why the research is important.
Reviewer 2 Report
The manuscript presents information that is of interest to the readership of plants. However, the MS has some grammatical errors. The authors have given the methodology and results regarding the estimation of Seed protein, seed oil content, and yield, however, the MS lacks discussion of these parameters. Moreover, the quality of the figures is very low as they are very blurry. The quality of the figures needs to modify. The tables too seem pasted as figures as the table presentation is very irregular. Botanical names mentioned in the MS must be italicized. The authors are requested to thoroughly check the whole MS for the correction of botanical names which must be italicized in the MS body and references.
Author Response
We thank the Reviewer for taking the time to review our manuscript and providing thoughtful and diligent comments. The Reviewer draws attention to the low quality of the figures which present as blurry upon close inspection. The figures have been adjusted to higher resolution. The Reviewer also makes note of the irregular formatting for the tables. We agree that the table presentation was very irregular, particularly for Table 1. The presentation of Table 1 was an attempt at visually representing where the data in each column/row came from, but as it appears it was not received the way we had anticipated. Table 1 has been completely restructured to include only the most important data discussed in the manuscript, and excluding the underlying data used in the average protein/oil/yield calculations. Table 2 has also been redesigned for easier readability. The Reviewer draws attention to the lack of italicization across botanical names. All botanical names have been corrected to be italicized, including the manuscript body and references. We value the comments and feedback offered by the Reviewer and have taken it into consideration as we edited the manuscript.
Reviewer 3 Report
The manuscript was well written and was a joy to read. Results were well presented and were supported by the data. The only comment is that if GmSWEET29 and GmSWEET34 are AtSWEET2 homologs and AtSWEET2 is expressed in roots and limits the efflux of sugar from At roots, then is the expression of GmSWEET29 or GmSWEET34 higher in root than leaf tissue. Also, if AtSWEET2 limits sugar efflux from the root to the rhizosphere, then was there anything qualitative about the roots ie, were there more/less nodules than the same strain in the east? Also, is nematode and other below ground pests pressure greater in the west compared to the east? Limiting sugar efflux might moderate that pressure assuming the sugar efflux is an attractant.
Since AtSWEET2 is expressed in the roots, I think you should explain your reasoning for only sampling leaf tissue.
Author Response
We thank the Reviewer for their review of our manuscript and appreciate the thoughtful and inquisitive feedback offered. The Reviewer inquired about the expression of GmSWEET29 and GmSWEET34 in root and leaf tissue in soybean. Because we lack expression data from root tissue for this study, we used a publicly available soybean expression atlas, The Bio-Analytic Resource for Plant Biology (BAR; http://bar.utoronto.ca/) ePlant Soybean eFP Viewer, a discussion of the tissue-specific expression patterns for GmSWEET29 and GmSWEET34 has been added to Discussion section 3.1. The Reviewer further poses questions regarding the relationship to the surrounding microbiome, both friend and foe. While we lack the soil microbiota data for the plots in this study, we feel this would make for an impactful future study. We have added an address to these biotic factors to Discussion section 3.1. The Reviewer suggests justifying our choice of leaf tissue for this study; we have added an address to Methods section 4.3. We have carefully considered the feedback from the Reviewer and appreciate their suggestions.
Round 2
Reviewer 1 Report
Abstract: This must be expanded to include the salient findings of the research carried out by the authors. Only passing remarks are not sufficient.
Lines 29-32: Genes' names should be made italics (Throughout the manuscript).
Line 133: What do you mean by 10 varieties (names)? In table 1, it is mentioned as Lines 1-10. What are the Lines 1-10 (Table 1)? Better mention the name of the variety rather than mentioning lines 1-10. Table 1 and the text mentioning the details of these lines need correction throughout the manuscript.
Lines 133-146: Mentioning '4 years and 4 locations' again and again for different varieties looks unwarranted and confusing. Authors must clarify this once, and then can discuss the results for different varieties/genotypes. Do this correction throughout the manuscript.
Line 177: Meaning not clear (what you mean by 'at least 36')?
Line 217: better use 'p-value ≤ 0.05'
in place of ' p-value < 0.05 '. Correct his throughout the manuscript.
Line 362: Use of 'we' can be avoided and sentences can be presented in a better way like 'The result showed an increase in the expression ......
Line 517: What is '00' ?
Line 517-523: Poorly written. Need to be presented in a better and more understandable way. Clarity is also required about the use of the term 'variety'. Are all the 10 lines released varieties (in Canada or other countries)? If not, then it will be better to use the term 'genotypes' over variety. Use of the word 'line(s)' for the genotypes/varieties should be avoided for clarity (throughout the manuscript).
Line 537: What is R5?
Line 541: Never start any sentence with any numerical value. Correct it. (better write, 'A total of 100....')
Overall, the manuscript still needs major English editing for the correct and meaningful flow of the sentences (for all the sections).
Author Response
The authors sincerely thank the Reviewer for their dedication and diligent review of our manuscript. All suggested changed have been made, including appropriate referencing, terminology, and clarification of text. The English presented has been carefully combed for accuracy, clarity, and flow. We sincerely appreciate the Reviewer for taking the time to read our manuscript and suggest improvements to make the paper stronger.